# Microbial Synthesis of Lactic Acid from Cotton Stalk for Polylactic Acid Production

**DOI:** 10.3390/microorganisms11081931

**Published:** 2023-07-28

**Authors:** Meenakshi Paswan, Sudipto Adhikary, Heba Hassan Salama, Alexandru Vasile Rusu, Antonio Zuorro, Bharatkumar Z. Dholakiya, Monica Trif, Sourish Bhattacharya

**Affiliations:** 1Department of Chemistry, Sardar Vallabhbhai National Institute of Technology, Surat 395007, India; meenakshipaswan2411@gmail.com (M.P.); bzdholakiya@chem.svnit.ac.in (B.Z.D.); 2Process Design and Engineering Cell, CSIR-Central Salt and Marine Chemicals Research Institute, Bhavnagar 346002, India; sudiptoadhikary827@gmail.com; 3National Research Centre, Dairy Department, Food Industries and Nutrition Research Institute, 33 El-Buhouth Str. (Former El-Tahrir Str.), Dokki, Giza 12622, Egypt; hebasalama11@yahoo.com; 4CENCIRA Agrofood Research and Innovation Centre, Ion Meșter 6, 400650 Cluj-Napoca, Romania; 5Department of Chemical Engineering, Materials and Environment, Sapienza University of Rome, 00184 Rome, Italy; antonio.zuorro@uniroma1.it; 6Food Research Department, Centre for Innovative Process Engineering (CENTIV) GmbH, 28857 Syke, Germany; monica_trif@hotmail.com; 7Academy of Scientific and Innovative Research (AcSIR), Bhavnagar 364002, India

**Keywords:** *Lactobacillus*, cotton stalk, cellulose, lactic acid, polylactic acid, *Lactococcus cremoris*, biomass valorization

## Abstract

Cotton stalk, a waste product in agriculture, serves as a beneficial, low-cost material as a medium for microbial synthesis of lactic acid as desired for polylactic acid synthesis. Cotton stalk was used as a substrate for microbial lactic acid synthesis, and a novel strain of *Lactococcus cremoris* was reported to synthesize 51.4 g/L lactic acid using cellulose recovered from the cotton stalk. In total, 18 Lactobacillus isolates were isolated from kitchen waste, soil, sugarcane waste, and raw milk samples screened for maximum lactic acid production. It was found that one of the *Lactococcus cremoris* isolates was found to synthesize maximum lactic acid at a concentration of 51.4 g/L lactic acid in the hydrolysate prepared from cotton stalk. The upstream process parameters included 10% inoculum size, hydrolysate containing reducing sugars 74.23 g/L, temperature 37 °C, agitation 220 rpm, production age 24 h. Only the racemic (50:50) mixture of D-LA and L-LA (i.e., D/L-LA) is produced during the chemical synthesis of lactic acid, which is undesirable for the food, beverage, pharmaceutical, and biomedical industries because only the L-form is digestible and is not suitable for biopolymer, i.e., PLA-based industry where high optically purified lactic acid is required. Furthermore, polylactic acid was synthesized through direct polycondensation methods using various catalysts such as chitosan, YSZ, and Sb_2_O_3_. PLA is biocompatible and biodegradable in nature (its blends and biocomposites), supporting a low-carbon and circular bioeconomy.

## 1. Introduction

The agricultural sector is one of the main sectors generating the largest quantities of agricultural solid waste, which may be allowed to accumulate indiscriminately and constitute a nuisance to global health and threat to food security or can be used as raw materials for bio-economy [1,2]. Cotton stalks are the residue remaining after cotton production, and are estimated to amount to 3.5–5 tons per ton of cotton crop. Global cotton production amounted to 25.8 million tons in 2018 and is expected to reach approximately 29.2 million tons by 2028.

Cotton waste accounts for 25% of total textile waste globally. Its utilization, complying with the demands of a circular economy, can solve a problem especially in Asia, including India, but also in Europe and particularly Poland. Poland itself generates over 236,661 Mg of cotton waste every year, which is clearly a massive amount to treat [3].

Efforts are being made in the textile industry to reduce cotton waste and promote recycling and upcycling practices to minimize the environmental impact associated with its disposal. Various techniques are being employed to make the most of cotton waste and create a more sustainable approach to cotton production and consumption. However, there is huge potential to utilize this cotton stalk material for production of biodegradable bioplastics through a circular economy model [4,5].

Cotton cellulose has various advantages over commonly used lignocellulosic material. One of the main advantages is its resistance to the action of acids and alkali compared with other type of cellulose [6,7,8,9]. Furthermore, cotton stalk can be easily subjected to hydrothermal conditions as well as alkaline catalysts for conversion of cellulosic material to reducing sugars which can be easily converted to lactic acid in the desired form [8,9,10,11,12,13]. However, appropriate pre-treatment processing can effectively increase the availability of reducing sugars for lactic acid fermentation [14,15]. Also, the changes in the community structure of the rhizosphere microbiome, especially the rhizosphere bacterial community structure in different soil, affect the chemical composition of cotton stalk. Six bacterial phyla were found dominant in the cotton rhizosphere bacterial community, including *Acidobacteria*, *Actinobacteria*, *Bacteroidetes*, *Planctomycetes*, *Proteobacteria*, and *Verrucomicrobia* [16].

Lactic acid (LA) is a valuable platform chemical with wide-ranging applications [17]. It is commonly used in the food, pharmaceutical, cosmetic, and textile industries [18]. For instance, it is used as preservative and pH-adjusting agent in the food industry. Recently, LA has also gained interest as a precursor of biodegradable polymer (PLA). Due to the wide variety of applications, the market value of LA increased to 2.7 billion dollars in 2020 and is predicted to grow at an annual rate of 8.0% from 2021 to 2028 [17]. Although LA can be produced via chemical processes, the fermentative process is preferred because it produces pure optical enantiomers. However, in most cases the fermentation media require the supplementation of nutrients to reach higher yields and good microbial growth. Many researchers have supplemented the medium with glucose and yeast extract (YE), which raises costs. This is a significant drawback when the process is to be implemented on an industrial scale for conversion of lactic acid to PLA [19].

Moreover, for effective conversion of LA to PLA, various catalysts such as antimony trioxide, zirconium, chitosan, and tin oxide for high molecular weight PLA have been reported. It has been shown than antimony trioxide is the preferred polycondensation catalyst because it offers high catalytic activity in the esterification reaction, resulting in high molecular weight PLA [20,21]. Yttrium-stabilized zirconium catalyst is a heterogeneous catalyst and percentage doping of Yttrium onto the zirconium is 3% mol. by weight, and is generally being used for this reaction. The benefit of this catalyst is its non-solubility in a solvent; it can be easily removed via the filtration method and thus avoids the presence of catalyst impurity in the final PLA product [22]. Researchers have also reported utilizing starch-based substrates involving corn, barley, or rice; however, these substrates require pre-treatment in the form of gelatinization, liquefaction, and saccharification before fermentation which is a cost-intensive process [23]. Other researchers have also tried simultaneous saccharification and LA production from cellulose wherein in situ enzymatic hydrolysis was attempted utilizing *Rhizopus* sp. or *Bacillus smithii* [24,25].

The aim of this study is to isolate LAB from natural sources, including waste materials, and then screen the isolates using CaCO_3_ as a neutralizing agent to identify potential hydrolysates that can further utilize the sugars present in lignocellulosic materials like cotton stalk for LA production. The potential microbe was screened and selected based on its homofermentative ability and then used in the synthesis of LA at the lab scale for polylactic acid synthesis using various catalysts such as chitosan, yttrium-stabilized zirconium (YSZ), and antimony trioxide, while optimizing other process parameters such as temperature, reaction time, and solvent (m-xylene).

## 2. Materials and Methods

### 2.1. Materials

De Man, Rogosa, and *Lactobacilli* (MRS) agar and broth, and chitosan were purchased from HiMedia, Mumbai, India. Glucose, yeast extract, sodium acetate, dipotassium hydrogen phosphate (K_2_HPO_4_), potassium dihydrogen phosphate (KH_2_PO_4_), magnesium sulfate (MgSO_4_), manganese sulfate (MnSO_4_), ferrous sulfate (FeSO_4_), calcium carbonate (CaCO_3_), and antimony trioxide (Sb_2_O_3_) were procured from Loba Chemie. N-butanol was purchased from MERCK Life Science Pvt. Ltd., Bangalore, India. LA was purchased from Rankem laboratory reagent, Gurgaon, Haryana, India. Distilled water was used at each step. Commercial PLA was purchased from Balson Industries, Pune, India. All remaining chemicals and solvents used for analysis were of analytical grade depending on their utilizations.

### 2.2. Experimental Section

De Man, Rogosa, and *Lactobacilli* (MRS) broth (Himedia, Mumbai, India) and MRS agar (Himedia, Mumbai, India) were used to culture LA, producing LAB isolated from natural sources.

Cultures were incubated in an incubator shaker (12–16 h) at 37 °C by inoculating from an isolated colony. However, 1% inoculum of pure culture stocks or culture was used during the initial experiments. After centrifugation, cell pellets from master cultures were re-suspended in milk-based freezing media (11% non-fat dry milk powder, 1% glucose, and 0.2% yeast extract) and kept frozen at 80 °C.

### 2.3. Isolation of LA Producing LAB

Samples were collected from different origins, which include kitchen waste collected from the CSMCRI departmental canteen, the sugar bagasse was collected from a local juice shop, the fresh cow milk (without preservatives) was collected from a local dairy farm in the morning, and soil was collected from the area of a nearby dairy farm in Bhavnagar, India (Figure 1 and Table 1). Sampling was carried out as per reports by Li et al., 2020 and Sardar et al., 2021 [26,27,28]. After collection of the raw samples, they were stored at 4 °C until used. Thereafter physico-chemical parameters (latitude and longitude, pH, temperature, TDS, BOD, and COD) were analyzed. The pH, temperature, and TDS of the samples were measured using an Elite PCTS meter (Thermofisher). For B.O.D., samples were collected in D.O. bottles and incubated at 20 °C in a B.O.D. incubator for five days. The biochemical oxygen demand (B.O.D.) was determined using a YSI ProOBOD probe. Chemical oxygen demand (COD) of the samples were tested as per ASTM D1252-95, 00(B) photometric method.

Isolation of lactic acid-producing bacteria was carried out in MRS medium. Enrichment of 10-fold diluted MRS broth was carried out in Whirl-Pak^®^ filter bags for 24 h (or 4 h) at 30 °C. Serial dilutions of enriched broth were streaked onto MRS agar plates according to the 5-sector method, and were buffered (pH 7.4 with 0.1 M sodium phosphate dibasic and 8 mM sodium phosphate monobasic), then incubated at 37 °C for 12 to 48 h. Bacterial isolates were subjected to submerged fermentation using the reported production medium to check the lactic acid production.

### 2.4. Morphological Characterization

Phenotypic identification of isolated bacterial strains was performed via examination of morphological, cultural, and biochemical characteristics, according to Bergey’s Manual of Systematic Bacteriology.

### 2.5. Fermentative Production of Lactic Acid and Screening of Isolates for Identification of Potential Isolates with Maximum Lactic Acid Production

CaCO_3_ has been considered as a neutralizing agent in LA production to reduce acid stress to the bacteria. Grown bacteria from the MRS agar medium were streaked on a modified deMan Rogosa Sharpe (MRS) agar medium supplemented with different concentrations of CaCO_3_ (0.5%, 1.5%, 2%, 2.5%, 3%, 3.5%, 4%, 4.5%, and 5%) and incubated at 37 °C. Colonies showing transparent halo surroundings were selected and processed for pure culture on MRS agar.

Inoculum medium (MRS broth) was inoculated with individual isolates and incubated at 37 °C for 24 h in an incubator shaker at 150 rpm under aerobic conditions. The production media consisted of glucose 50 g/L, yeast extract 30 g/L, dipotassium hydrogen phosphate 0.5 g/L, potassium dihydrogen phosphate 0.5 g/L, magnesium sulfate 0.6 g/L, sodium acetate 1 g/L, manganese sulfate 0.03 g/L, and ferrous sulfate 0.03 g/L in a 1000 mL conical flask. The production medium was inoculated with 10% (*v*/*v*) inoculated medium and incubated at 37 °C for different incubation times (24 h, 48 h, 72 h, and 96 h). The production medium was centrifuged at 10,000 rpm for 10 min, after incubation. An HPLC instrument was used for determining produced LA [22], using the UV spectrophotometric method at 390 nm [29].

The potential isolated LABs were further subjected to sub-culturing 3–4 times. The quadrant streak method was used for maintenance of the cultures and potential isolates of LABs confirming maximum LA production were kept on MRS agar slants and plates (both) at 4 °C and streaked every 15–20 days. Prior to use, LAB strains were activated in MRS broth at 37 °C for 24 h and sub-cultured in MRS agar at 37 °C for 24–48 h, then maintained in 20% glycerol for storage at −20 °C for future use.

### 2.6. Molecular Characterization of Potential LAB Isolates

Using 16S rRNA, potent bacterial strains were identified. DNA was extracted from a single colony of isolates using a Gene Jet genomic DNA purification kit protocol (Thermofisher, Mumbai, India). The universal primers 27F and 1429R (forward and reverse primers) were used, respectively, in the PCR. For amplification of 16S rDNA bacterial sequences, 27F (forward) and1492R (reversed) primers are listed in Table 2. Following PCR, the generated DNA sequence was compared using the BLAST (basic local alignment search tool) programme on the NCBI website, which can be accessed at http://www.ncbi.nlm.nih.gov (accessed on 20 November 2022). The resulting DNA sequence was identified and submitted to GenBank through comparisons and led to construction of a phylogenetic tree.

### 2.7. LA and Growth Production Profiles

Based on the screening studies for LA production, eight LAB isolates were selected for a time–course study to generate their growth and LA production profiles. For the growth study, MRS broth medium was prepared for the selected strains in 250 mL Erlenmeyer flasks and sterilized by autoclaving at 121 °C for 15 min. After cooling, isolated strains were added to the broth with 0.5% CaCO_3_ and incubated for 54 h. Growth kinetics were studied through estimating optical density every 6 h interval using a spectrophotometer at 600 nm.

For LA profiles, 600 mL of production medium in a 1000 mL flask was prepared and sterilized. After 12 h, the production medium inoculated with 60 mL of 18 h old culture had a LAB count of 9.6 log CFU/mg at OD 600 nm and was kept at 37 °C at 100 rpm in an incubator shaker. However, in every 6 h interval till 54 h, 10 mL samples were collected for estimating LA concentration using a UV spectrophotometer at 390 nm and the FeCl_3_ method. During each sampling time, 10 mL of the cultures were removed aseptically from each flask and assayed for DNSA (sugar concentration), pH, LA, and biomass (cell dry weight). Using optical density measurements at 600 nm, biomass concentration was also quantified [30,31].

### 2.8. Cellulose Recovery from Cotton Stalk

The cotton stalk material was powdered to 200 mesh size and processed for cellulose recovery. The powdered material was dried and bone-dry material was added to 500 mL of 2% NaClO_2_ solution, and pH was adjusted to 4.0 using HCl. The whole content was allowed to bleach at 65 °C for 4 h and washed. Following washing, 250 mL of 5% HCl was added to the bleached slurry, heated till boiling, and kept at room temperature overnight. The slurry was washed to neutrality and dried to obtain cellulose after the incubation.

### 2.9. Saccharification of Recovered Cellulose for Its Effective Conversion to Hydrolysate Containing Reducing Sugars

In triplicate, 1 g of cellulose was added to 20 mL of 51% H_2_SO_4_ and hydrolyzed at 45 °C for 4 h with continuous stirring in a 100 mL reactor connected to a condenser at the top so that the total volume was maintained. Then, the slurry was centrifuged at 10,000 rpm at ambient temperature to obtain the supernatant containing sugars, which were further neutralized with alkali (NaOH) for removing the acid and subjected to centrifugation.

### 2.10. Microbial Synthesis of LA

The obtained supernatant containing sugars was concentrated to 10 mL volume (10 mL DI water was evaporated in a rotary evaporator under vacuum) and was used as the carbon source for submerged fermentation for production of LA using the isolated potential strain. The inoculum size was considered as 10% of 18 h old culture having a LAB count of 9.6 log CFU/mg at OD 600 nm.

### 2.11. Purification of LA

A traditional process for purifying LA includes precipitating with calcium lactate, esterification, and hydrolysis using reactive distillation. It is a cost-effective, easy, and reliable process, but it produces a significant amount of CaSO_4_, which is recognized as an environmental contaminant [32]. For this reason, an alternative and ecologically more accurate process was used for purification based on a reactive extrusion method where an amine compound, along with a solvent, is used for LA recovery [33]. A 10 mL sample of fermented broth was taken and 5 g of ammonium sulfate added and thoroughly dissolved. Before keeping for 2 h at 30 °C, 30 mL of n-butanol was added and vigorously vortexed for adequate mixing. The mixture was then poured into a separating funnel and let stand until the aqueous and organic phases could be identified. The top organic phase was recovered following phase separation. In a rotavapor, the collected organic phase was vaporized at 50 °C. The final product was slightly oily in texture. The LA in the dried organic phase was collected, dissolved in 5 mL Milli Q water, and measured using UV spectrophotometric methods [26].

### 2.12. Synthesis of PLA from LA

PLA can be synthesized by the direct polycondensation method from microbial synthesis of LA using the potential isolate with maximum production. Polymerization of LA using different catalysts i.e., Sb_2_O_3_, chitosan, and 3%mol YSZ was performed. In the direct polycondensation method, 181 g lactic acid, catalyst, and 150 mL m-xylene were added into a 3-neck round-bottom flask (500 mL) equipped with mechanical stirrer, Dean and Stark assembly, and thermometer. With help of the Dean and Stark assembly, water was azeotropically distilled off at elevated temperature. The reaction was carried out at 160 °C for 24 h under nitrogen gas flow and the temperature raised to 200 °C for half an hour. The resultant polymer was of dark brown color and high viscous in nature. The reaction mass was cooled to room temperature and the resulting polymer was dissolved in solvent such as chloroform and thereafter precipitated twice in excess methanol. The precipitated product was filtered and dried in a vacuum oven at 60 °C for 3 h for complete removal of bounded and unbounded solvent. All the prepared solutions were poured into Petri dishes and dried for 12 h at 60–100 °C. After drying, the films were peeled off and placed in a desiccator at room temperature for further analytical characterization and the resultant polymers were labelled as PLA-Sb_2_O_3_, PLA-chitosan, and PLA-YSZ, respectively.

### 2.13. Analytical Characterization

#### 2.13.1. Quantitative Estimation of LA Using Spectrophotometric Method

A 100 mL volumetric flask containing 0.3 g of iron (III) chloride was taken, diluted with water, and stirred until the salt was completely dissolved. The solution was kept at room temperature of 25 ± 5 °C. For calibration, the calibrating curve of the LA was prepared using standard LA. Fermentation broth was taken after centrifugation. A 50 μL fermentation broth of respective isolate was added to 2 mL of a 0.2% solution of iron (III) chloride and stirred. At 390 nm, the absorbance of the colored solutions that were obtained was measured. In the process, 2 mL of iron (III) chloride solution (0.2% concentration) was used as a reference solution. While the color solution was stable for 15 min, the absorbance of colored solutions depended on the amount of LA present in the broth. LA concentration in the fermentation broth was determined via calibration [29] using a Shimadzu UV-visible spectrophotometer UV-1603 (Shimadzu Analytical (India) Pvt. Ltd., Mumbai, India).

#### 2.13.2. Determination of LA through HPLC

At 10,000 rpm, fermentation broth was centrifuged for 10 min at room temperature, and with the help of 0.2 µm syringe filters, supernatant was filtered for LA estimation by HPLC. The LA was determined using a Shimadzu HPLC system equipped with (Rezex^TM^ ROA-Organic acid H+) LC column (7.8 × 300 mm) and refractive index detector (RID) where 5 mM H_2_SO_4_ was used as a mobile phase at a flow rate of 0.6 mL/min.

#### 2.13.3. FTIR Analysis

FT-IR spectra were recorded 45 times using a Shimadzu FTIR-8400S from the range of 4000 to 400 cm^−1^. The samples were prepared by taking small piece of each and uniformly mixing with KBr powder. The software Labsolutions IRsolution was used for analysis and spectra were taken in transmittance mode. FT-IR spectra of the synthesized LA, fermentation broth, and resultant polymers such as PLA-Sb_2_O_3_, PLA-chitosan, and PLA-YSZ were recorded and compared with standard LA and commercial PLA, respectively.

#### 2.13.4. Nuclear Magnetic Resonance (NMR) Analysis

NMR (both ^1^H and ^13^C) spectra were recorded with a JNM-ECZS NMR spectrometer (400 MHz) for standard LA, synthesized LA, and resultant polymers such as PLA-Sb_2_O_3_, PLA-YSZ, and PLA-chitosan, where CDCl_3_ (for resultant polymers) and D2O (for synthesized LA) were used as solvents and tetramethyl silane (TMS) was used as internal reference.

#### 2.13.5. Gel Permeation Chromatography Analysis of Resultant Polymers

Gel permeation chromatography (GPC) was used for molecular weight determination of resultant polymers such as PLA-Sb_2_O_3_, PLA-YSZ, and PLA-chitosan. According to the GPC spectrum, the MWn, MWw, and polydispersity index (PI = MWw/MWn) were evaluated. The GPC unit was a PerkinElmer USA series, Turbo matrix-40 model. Columns were calibrated with polystyrene standards, and tetrahydrofuran was used as both solvent and eluent.

### 2.14. Statistical Analysis

Data were analyzed using OriginPro 19. Significance testing and Pearson correlation analyses were conducted with SPSS 21.0. R was used for performance redundancy analysis and principal component analysis.

## 3. Results

### 3.1. Isolation and Characterisation of LAB

A CaCO_3_ dissolution helped to identify LAB on the MRS agar medium supplemented with 0.5% CaCO_3_. The results include a total of 16 bacterial isolates from raw milk samples, soil samples, and natural waste such as kitchen waste and sugarcane waste. The resulting dissolution of CaCO_3_ and colony morphology are presented in Table 3. Upon preliminary diagnosis, 8 out of 16 LAB isolates were Gram-positive and capable of fermenting glucose and releasing acid in media, as shown in Table 3.

After incubation at different temperatures, a few LAB strains were grown for 24 h and a few strains were grown for up to 48 h at 37 °C. At 45 °C, no growth was found on MRS agar media, while at 30 °C, few colonies were found. Colonies from 37 °C and 30 °C were selected for LA production. Presumptive LABs were isolated on MRS agar media and screened for high LA production on MRS agar media supplemented with various concentrations of CaCO_3_. It was found that some isolates at 0.5% CaCO_3_ were showing a clear zone, i.e., dissolution of CaCO_3_ on agar plates, while at higher concentrations of CaCO_3_, isolates were not able to produce LA which may be due to excess calcium carbonate presence in the MRS agar media, which can be toxic for isolates (Figure 2). Because of the strong ambient osmotic pressure produced by the high CaCO_3_ concentrations, LA production was hindered due to growth-associated metabolism [34,35,36]. However, Dosuky et al., 2022 screened lactic acid-producing isolates using salted cheese whey and whey permeate mixture [34]. It was found that one of the potential isolates Ent. 58 was able to produce lactic acid at a concentration 16.23 ± 0.03 g/L. Also, no one has utilized calcium carbonate in the medium for lactic acid production. Yang et al., 2015 reported that using calcium carbonate, lactic acid production can be increase, however, they used 0.5% calcium carbonate along with MRS medium containing glucose as a carbon source [32]. Also, it was reported by Yang et al., 2015 [36] that 0.5% calcium carbonate was the optimum concentration for optimum lactic acid production. In that regard, our results also showed a similar pattern and were in line with reports wherein maximum lactic acid production on the plates was found with 0.5% calcium carbonate. However, in our case, we have developed the protocol on the plates so that there is no need for screening in the shake flasks which is carried out in most of the reports. Also, this may be considered a rapid and cost-effective screening technique wherein a potential lactic acid-producing isolate can be identified in a single attempt.

Those isolates showing a clear zone were selected for further genotypic and phenotypic analysis, and Gram staining was performed. On Gram staining, Gram positive samples were selected as LAB. Furthermore, morphological, physiological, and biochemical characteristics of those isolates with respect to standard descriptions given in Bergey’s *Manual* [36] are compared and represented in Table 3. It was found that the lactic acid production was optimum in the medium containing 0.5% CaCO_3_, which was also reported by Yang et al., 2015 [34]. In that regard, in order to screen potential lactic acid-producing bacteria, isolates were streaked in MRS medium containing 0.5% CaCO_3_, 1% CaCO_3_, 2% CaCO_3_, 3% CaCO_3_, 4% CaCO_3_, and 5% CaCO_3_. MRS medium with 0.05% CaCO_3_ and MRS medium without CaCO_3_ were kept as positive and negative controls. It was found that MRS medium with 0.5% CaCO_3_ was the optimum medium concentration for screening of potential isolates for maximum lactic acid production. Also, if we increased the calcium carbonate concentration, the growth of the bacteria could not be observed on the plate. So, based on this protocol, we were able to identify a potential strain of *Lactococcus cremoris* which may have maximum lactic acid production. Furthermore, the lactic acid production of the *Lactococcus cremoris* strain was checked in the reported medium and it was found that it can produce 51.4 g/L lactic acid in submerged fermentation.

### 3.2. Molecular Identification of Potential LAB Isolate Having Maximum LA Production

The amplified DNA was subjected to 16s rRNA sequencing using Sanger sequencing and the obtained sequence was subjected to nucleotide blasting at NCBI, which confirmed *Lactococcus cremoris* (740 bp) and *Lactococcus lactis* strain (1036 bp). *Lactococcus cremoris* and *Lactococcus lactis* are species of LAB commonly used in the dairy industry for the production of fermented dairy products such as cheese and yogurt. They are known for their ability to produce LA through the fermentation of lactose, the primary sugar in milk. The specific LA production by *Lactococcus cremoris* can vary depending on various factors such as the specific strain of bacteria, fermentation conditions, and the composition of the growth medium. The phylogenetic tree was prepared (Figure 3) and the nucleotide sequence was submitted to NCBI, with GenBank accession nos. ON972427.1 and ON205829.1, respectively.

### 3.3. Analytical Characterization

#### 3.3.1. Using HPLC Method for LA Production

The fermentation broth was centrifuged at 10,000 rpm for 10 min at room temperature, and using 0.2 µm syringe filters the supernatant was filtered for LA estimation by HPLC (Figure 4). The main end-product of LAB isolates is to produce LA by utilizing nutrient components. In these experiments, results showed that LAB-1 (Figure 4) and other strains produce LA in fermentation broth (refer to Appendix A).

#### 3.3.2. Using U.V Spectrophotometric Analysis for LA Determination

For LA determination, a calibration curve was plotted using different concentrations of LA in the range from 0 to 100 g/L. Here, the absorbance of iron (III) lactate solution is proportional to the concentration of LA. The correlation coefficient is 0.995 and confidence level 0.000001. Concentration of LA produced by isolates was calculated using a calibration curve (Table 4).

#### 3.3.3. LA Purification from Fermentation Broth

Downstream processing of LA produced by *Lactococcus cremoris* (LAB-1) was attempted using phase partitioning by using n-butanol as a single solvent. The *Lactococcus cremoris* (LAB-1) produced 0.51 M of LA in fermentation broth after 58 h of growth. The pH of the broth at the end of fermentation was 4.73. However, various parameters such as pH of fermentation broth, ammonium sulfate, and n-butanol concentration affect the purification of LA. The obtained yield and purity of LA was quite comparable to other studies. Compared with the different processes, the present method is more straightforward as LA was recovered in one-step phase partitioning involving n-butanol and ammonium sulfate. Furthermore, the n-butanol used was recovered during rotary evaporation and can be used again. Moreover, other procedures have their inherent drawbacks, such as membrane processes with the problem of membrane fouling, and calcium hydroxide precipitation having the disadvantage of generation of gypsum left over in large amounts. The methodology developed in this study is devoid of any such drawbacks and thus deemed simple, economical, and efficient (Figure 5).

#### 3.3.4. FTIR Analysis

The FTIR spectra of standard LA, microbial synthesized LA, and fermentation broth are depicted in Figure 6. The peaks of synthesized LA are similar to those present in standard LA (Figure 6A). In the microbial-synthesized LA, -OH bond stretching was found at wavenumber 3135.7 cm^−1^ indicating -OH groups. A strong C=O absorption was observed at 1746.6 cm^−1^ representing carbonyl stretching. A band responding to bending vibrations of C–H was observed at 1461.8 cm^−1^ in the spectrum. It was also observed that -CH and -CH_3_ bond stretching occurred at 2977.7 cm^−1^ wavelength in a molecule.

However, the FTIR spectra of the resultant polymers with different catalysts (PLA-YSZ, PLA-Sb_2_O_3,_ and PLA-chitosan) were found to be very similar to standard PLA (Figure 6B). The resultant polymers revealed characteristic absorption peaks of ester at 1770, 1749.5, and 1774.9 cm^−1^, respectively. As LA polymerizes, the hydroxyl group reacts with the carboxylic group of another molecule, which is observed at wavelength 3505.4, 3499.5, and 3422.9 cm^−1^ indicating -OH bond stretching [32]. As the numbers of hydroxyl groups were reduced, the sharper absorption peak of C=O stretching (1761 cm^−1^) was observed in the synthesized polymer (Figure 6B).

#### 3.3.5. NMR (Both ^1^H NMR &^13^C NMR) Analysis

The ^1^H NMR spectra of all resultant PLA (Figure 7) revealed the signal of methane proton resonances in the main chain at 5.15 ppm, as it is attached with an electronegative oxygen atom and the signal of methyl proton occurs in the main chain at δ(CH_3_) 1.5 ppm [36]. Generally, carbolic proton shifts appear above 10 ppm but sometimes in polymer they comes at around 4 to 7 ppm, while CDCL_3_ (solvent) peaks at 7.3 and TMS (internal reference) peaks at 0. The above results confirmed that PLA was formed. For PLA-Sb_2_O_3_, PLA-chitosan, and PLA-YSZ (Figure 8), the ^13^C NMR spectra were composed of lines located at δ(CH_3_) 16.9 ppm, δ(CH_4_) 68.98 ppm, and δ(CO) 169.5 ppm which correspond to the methyl, methane, and carbonyl carbon atoms, respectively.

#### 3.3.6. Identification of Molecular Weight of PLA Using Gel Permeation Chromatography (GPC) Analysis

The molecular weight of PLA was determined using GPC spectra (refer to Appendix A) and is shown in Table 5. However, the presence of impurities and moisture or water, reaction temperature, solvent amount, and catalyst impurities can affect the polymerization steps and can act as a hinderance in high-molecular-weight PLA formation [37]. However, during reaction, removing water is one of the essential factors. It is said that formation of water molecules during polymerization causes the bonds to break in chains, which affects the molecular weight. The higher the polymerization time and temperature, the greater the molecular weight will be [38]. The molecular weight of PLA can vary depending on its composition and polymerization process. Typically, the molecular weight (MW) of PLA ranges from several thousand to several hundred thousand grams per mole (g/mol) [39]. Out of the three catalysts, PLA-YSZ with a molecular weight around 3854 was obtained, and can be utilized in biomedical applications such as sutures, bandages, and drug-delivery applications where the L form of PLA is recommended.

### 3.4. Statistical Analysis

Multivariate regression analysis of the effect of different concentrations of calcium carbonate in the medium, the effect of acid hydrolysis, and the effect of reducing sugar concentration was carried out. It was found that the Pearson coefficient for the effect of CaCO_3_ and the effect of reducing sugars on LA production was 1. However, the Pearson coefficient for the effect of acid hydrolysis was 0.554, which shows moderate correlation. In other words, reducing sugars in the hydrolysate may be considered the limiting factor for microbial synthesis of LA in the present case.

## 4. Discussion

Cotton stalk is one of the agriculture residues whose effective disposal is still a challenge, as its disposal by incineration may lead to toxic gases, and generation of ash, which is harmful for the environment. In other words, cotton stalk is a waste material which can serve as highly useful and low-cost material for preparing strategic goods. Although cotton stalk may be considered lignocellulosic material, it contains a considerable amount of cellulose which can be used for fermentation. However, few reports have demonstrated the effect of the rhizosphere microbiome on the cotton stalk composition. The changes in the community structure of the rhizosphere microbiome, especially the rhizosphere bacterial community structure in different soil, affect the chemical composition of cotton stalk. Six bacterial phyla were found dominantly in the cotton rhizosphere bacterial community, including *Acidobacteria*, *Actinobacteria*, *Bacteroidetes*, *Planctomycetes*, *Proteobacteria*, and *Verrucomicrobia*, and their variation may affect the soil composition and also affects the cotton roots and their chemical composition, which can affect the composition of cotton stalk [16].

In the current context, efforts were made to produce LA from the cellulosic hydrolysate of cotton stalk wherein further LA was converted to PLA. In total, 18 *Lactobacillus* isolates were isolated from kitchen waste, soil, sugarcane waste, and raw milk samples screened for maximum lactic acid production. It was found that one of the *Lactococcus cremoris* isolates synthesized maximum lactic acid at a concentration of 51.4 g/L lactic acid in the hydrolysate prepared from cotton stalk. The upstream process parameters included 10% inoculum size, hydrolysate containing reducing sugars 74.23 g/L, temperature 37 °C, agitation 220 rpm, and production age 24 h. The hydrolysate was prepared from cotton stalk through chemical pretreatment wherein initially the cellulose was recovered from the hydrolysate and then subject to acid hydrolysis to obtain the hydrolysate containing reducing sugars. Further, the submerged fermentation was carried out utilizing 10% (*v*/*v*) inoculum of *Lactococcus cremoris* in the cellulosic hydrolysate (prepared from cotton stalk) at 37 °C for 48 h. Thereafter, after completion of fermentation, the broth was centrifuged to obtain the supernatant which was subjected for recovery of LA followed by PLA production from purified LA. Abedi and Hashemi, 2020 reported that using *Lactobacillus* strains may be considered to have great commercial importance due to their high acid tolerance, high yield, and productivity, and capacity to be engineered for the selective production of L/D-lactic acid. Amongst all reported *Lactobacillus* strains, *Lactobacillus delbruckii* NCIMB 8130 (90 g/L LA production), *Lactobacillus delbrueckii* sp. lactis ATCC 12315 (100 g/L LA production), and *Lactobacillus delbrueckii* sp. *bulgaricus* Ch H 2217 (115 g/L LA production) were found to have maximum lactic acid production [40,41,42,43,44,45]. However, with the lignocellulosic hydrolysate, maximum lactic acid production was reported using *Lactobacillus delbrueckii* sp. lactis 447 with 55 g/L lactic acid production with 85% carbon utilization efficiency. However, to date, lactic acid production from *Lactococcus cremoris* had not yet been reported. For the first time, *Lactococcus cremoris* has reported to producing lactic acid at a concentration of 54 g/L utilizing lignocellulosic (cotton stalk) hydrolysate.

## 5. Conclusions

Cotton stalk is an agricultural residue which raises serious environmental concerns, as its incineration leads to formation of toxic gases and ash content, thus contaminating the environment. The possible sustainable solution which also is the current need of the hour is to develop a sustainable solution for production of cost-effective biodegradable plastics such as PLA from cotton stalk. These can be produced from biopolymers obtained from biomass, such as starch, cellulose, or proteins. The developed process described in the manuscript demonstrates a waste-to-wealth strategy for microbial synthesis of LA which can be utilized for polylactic acid, which is a wonderful biodegradable bioplastic. Furthermore, PLA is considered to be biocompatible and biodegradable in nature (including its blends and bio-composites), supporting a low carbon and circular bioeconomy.

## Figures and Tables

**Figure 1 microorganisms-11-01931-f001:**
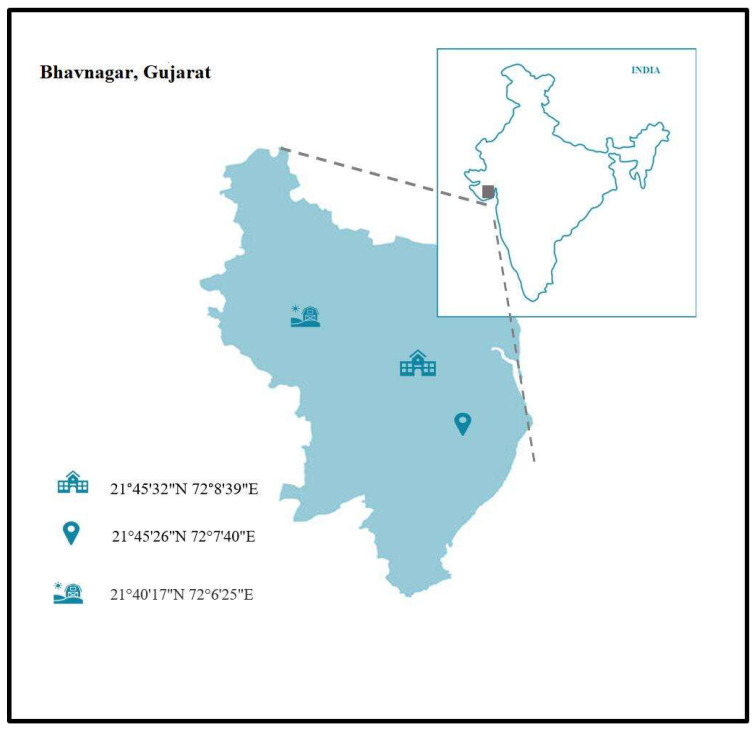
Geographical distribution of the sampling locations in Bhavnagar, Gujarat (India).

**Figure 2 microorganisms-11-01931-f002:**
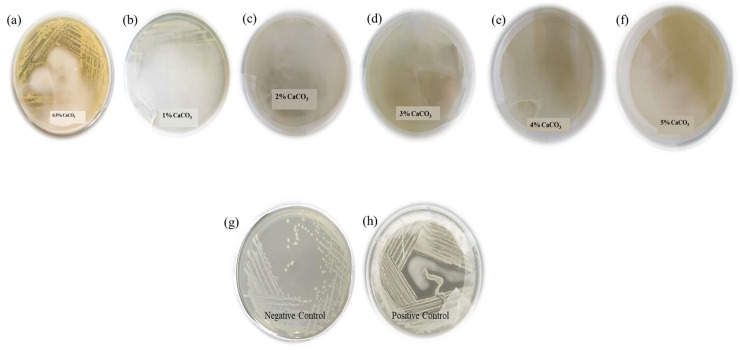
Effect of CaCO_3_ concentration on lactic acid production on plates streaked with (**a**) MRS media with 0.5% CaCO_3_; (**b**) MRS media with 1% CaCO_3_; (**c**) MRS media with 2% CaCO_3_; (**d**) MRS media with 3% CaCO_3_; (**e**) MRS media with 4% CaCO_3_; (**f**) MRS media with 5% CaCO_3_; (**g**) MRS media without CaCO_3_ (negative control); (**h**) MRS media with 0.05% CaCO_3_ (positive control).

**Figure 3 microorganisms-11-01931-f003:**
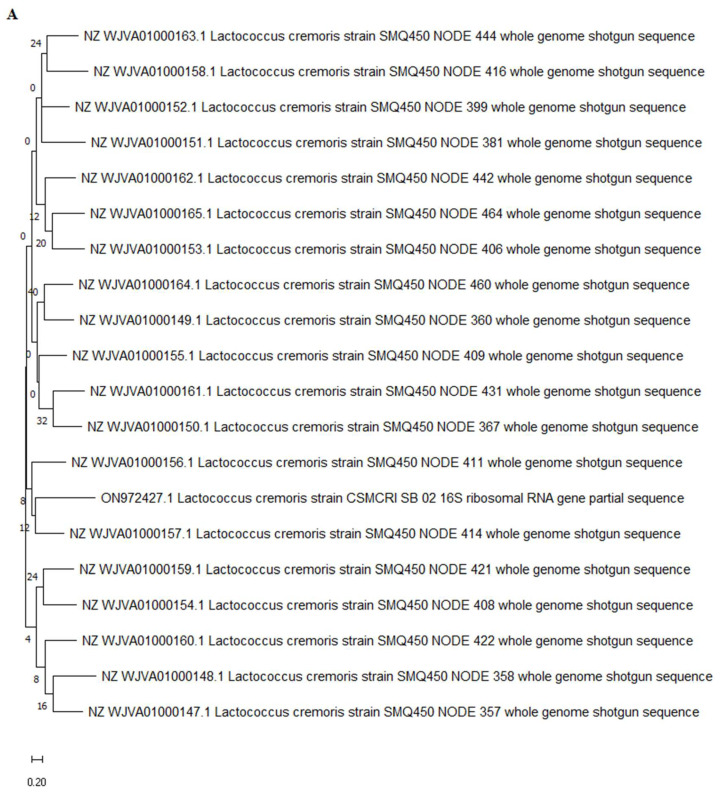
Phylogenetic tree of potential isolate (**A**) LAB-1 (*Lactococcus cremoris*) and (**B**) LAB-2 (*Lactococcus lactis*).

**Figure 4 microorganisms-11-01931-f004:**
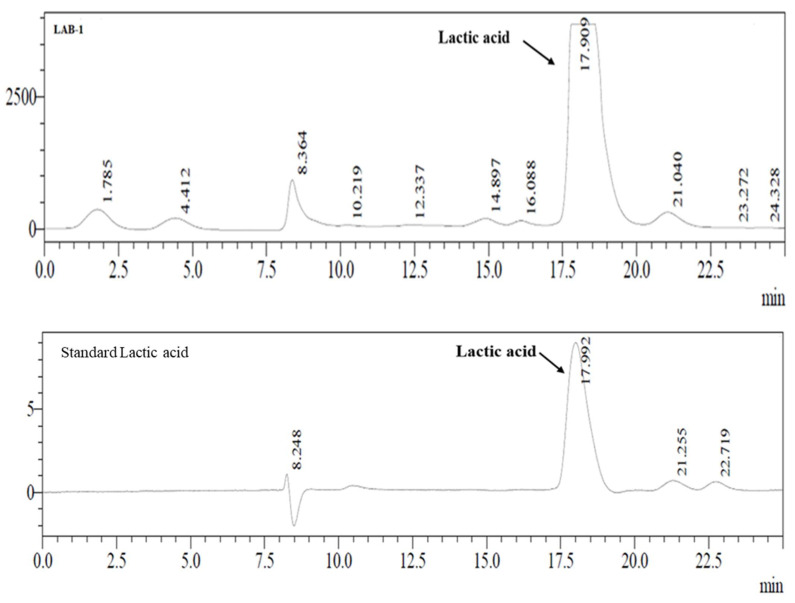
HPLC chromatograms of LA production from LAB-1 isolates and comparison with standard LA.

**Figure 5 microorganisms-11-01931-f005:**
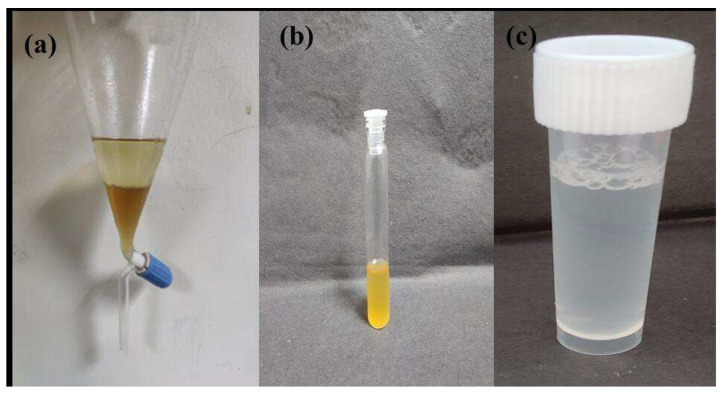
Purification of LA from fermentation broth: (**a**) phase partitioning method, (**b**) LA and (**c**) pure LA (after treatment).

**Figure 6 microorganisms-11-01931-f006:**
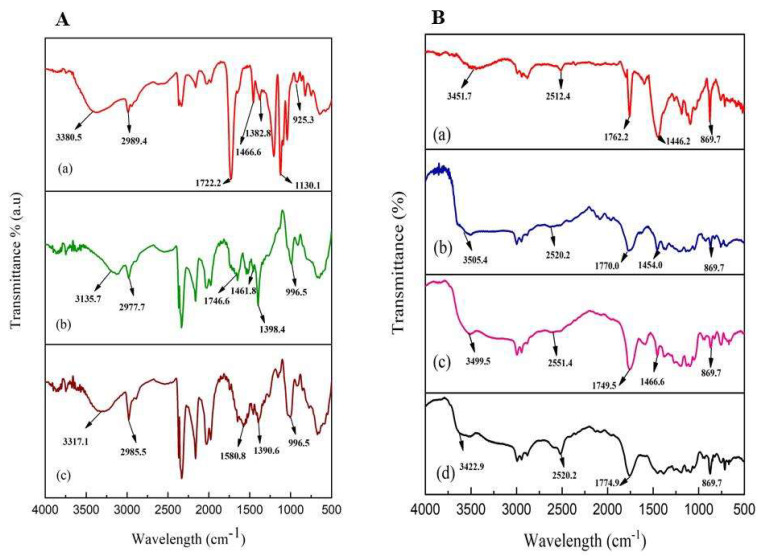
FTIR spectra (**A**) of (**a**) standard LA, (**b**) synthesized LA, and (**c**) fermentation broth, and (**B**) of (**a**) standard PLA, (**b**) PLA-YSZ, (**c**) PLA-Sb2O3, and (**d**) PLA-chitosan.

**Figure 7 microorganisms-11-01931-f007:**
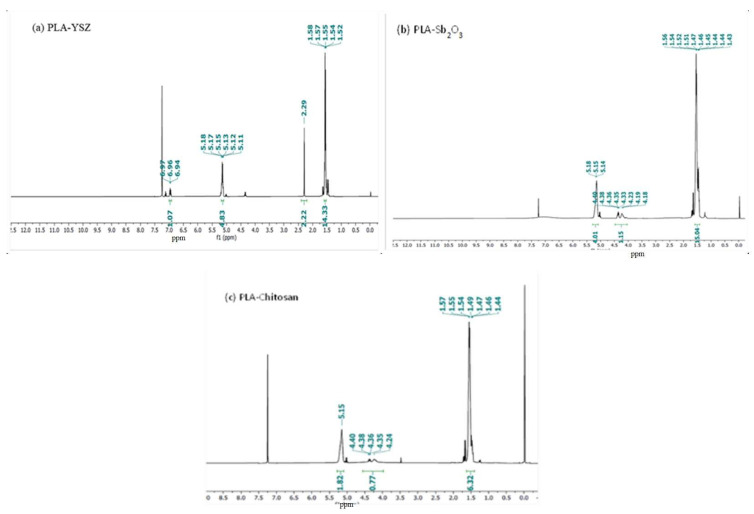
^1^H NMR of resultant polymer, i.e., (**a**) PLA-YSZ, (**b**) PLA-Sb_2_O_3,_ and (**c**) PLA-chitosan.

**Figure 8 microorganisms-11-01931-f008:**
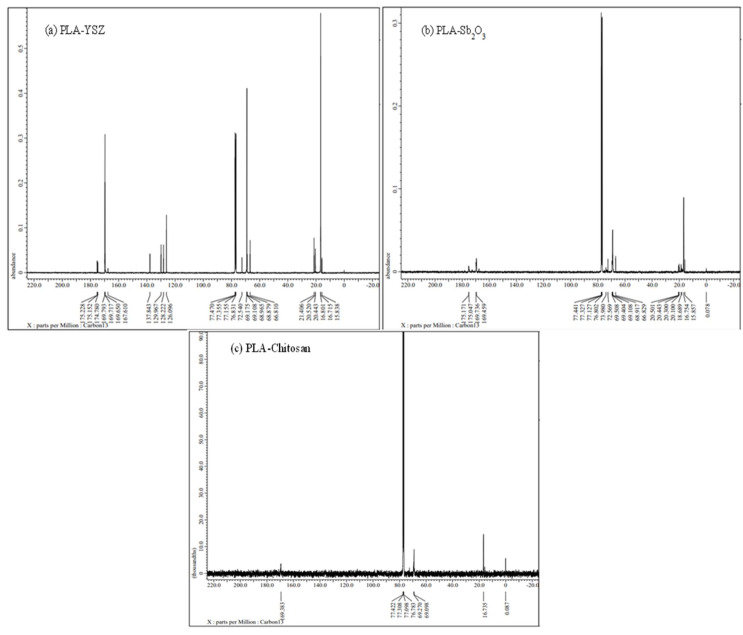
^13^C NMR of resultant polymer, i.e., (**a**) PLA-YSZ, (**b**) PLA-Sb_2_O_3,_ and (**c**) PLA-chitosan.

**Table 1 microorganisms-11-01931-t001:** Geographical locations of collection site and physicochemical parameters of samples collected.

Sample No.	Sampling Area	Latitude and Longitude	pH	Temperature	TDS (g/L)	BOD (g/L)	COD (g/L)
1.	Kitchen waste collected from CSMCRI departmental canteen	21°45′32″ N 72°8′39″ E	8.4	42	150.6	1.2	0.6
2.	Dairy farm, Bhavnagar	21°40′17″ N 72°6′25″ E	6.1	40	73	1.3	2.5
3.	Sugarcane bagasse	21°45′26″ N 72°7′40″ E	7.4	38	* BDL	* BDL	* BDL

* BDL—Below detectable limits.

**Table 2 microorganisms-11-01931-t002:** Primer (16S rDNA) used for molecular characterization.

Primer’s Name	Primer Sequence
27F (Forward)	5′-AGAGTTTGATCCTGGCTCAG-3′
1429R (Reversed)	5′-GGTTACCTTGTTACGACTT-3′

**Table 3 microorganisms-11-01931-t003:** Identification of selected LAB isolates for LA Production.

Isolates Name	Gram’s Staining	Growth	Source	Catalase Test	Dissolution of CaCO_3_	Colony Characterization
LAB-1	+ve	24 h, 37 °C	Raw cow milk	−ve	Yes	Medium circular size, smooth texture, glossy appearance, smooth edge, opaque, whitish colony
LAB-2	+ve	24 h, 37 °C	Raw cow milk	−ve	YES	Medium circular size, smooth edge, glossy appearance, opaque, whitish colony
LAB-3	+ve	24 h, 37 °C	Raw cow milk	−ve	YES	small circular size, smooth edge, rough appearance, translucent, pale white colony
LAB-4	+ve	24 h, 37 °C	Raw cow milk	−ve	YES	Medium circular size, smooth edge, glossy appearance, opaque, whitish colony
LAB-5	+ve	24 h, 37 °C	Raw cow milk	−ve	YES	Medium circular size, smooth edge, glossy appearance, translucent, whitish colony
LAB-6	+ve	24 h, 37 °C	Kitchen waste	−ve	YES	Medium circular size, smooth edge, opaque, whitish colony
LAB-7	+ve	24 h, 37 °C	Kitchen waste	−ve	YES	Small, circular size, smooth edge, opaque, whitish colony
LAB-8	−ve	48 h, 30 °C	Kitchen waste	−ve	No	Large, circular size, wavy edge, glossy appearance, opaque, whitish colony
LAB-9	+ve	24 h, 37 °C	Kitchen waste	−ve	No	Medium circular size, smooth edge, glossy appearance, opaque, whitish colony
LAB-10	+ve	24 h, 37 °C	Kitchen waste	−ve	No	Medium circular size, smooth edge, glossy appearance, opaque, whitish colony
LAB-11	−ve	24 h, 37 °C	Kitchen waste	−ve	No	Medium circular size, smooth edge, glossy appearance, opaque, whitish colony
LAB-12	+ve	24 h, 37 °C	Sugarcane bagasse	+ve	YES	Small, circular size, rough edge, glossy appearance, transparent pale-yellow colony
LAB-13	+ve	48 h, 37 °C	Sugarcane bagasse	−ve	NO	Medium circular size, smooth edge, glossy appearance, opaque, whitish colony
LAB-14	−ve	48 h, 30 °C	Sugarcane bagasse	+ve	NO	Medium circular size, smooth edge, rough appearance, opaque, whitish colony
LAB-15	−ve	24 h, 37 °C	Soil sample	+ve	NO	Small, circular size, smooth edge, glossy appearance, opaque, whitish colony
LAB-16	+ve	24 h, 30 °C	Soil sample	+ve	NO	Medium circular size, smooth edge, glossy appearance, opaque, whitish colony

**Table 4 microorganisms-11-01931-t004:** Determination of LA through U.V Spectrophotometric analysis.

Isolates	LA (gm/L^−1^)	Dry Weight Biomass (gm/mL)	Sugar Consumed (gm/L^−1^)
LAB 1	51.3564	0.0624	41.353
LAB 2	29.1798	0.0260	25.2038
LAB 3	40.4100	0.06385	38.243
LAB 4	36.908	0.06585	36.832
LAB 5	41.3456	0.05784	38.988
LAB 6	39.0456	0.07543	36.982
LAB 11	46.0685	0.06843	38.654
LAB 14	39.6854	0.05876	37.288

**Table 5 microorganisms-11-01931-t005:** Molecular weight determination of resultant polymer through gel permeation chromatography.

Sample Code	Catalyst (1%)	Time (h)	Temperature (°C)	MWn	MWw	PI
PLA-YSZ	1.0	24 h	180 °C	2805	3854	1.374
PLA-Chitosan	1.0	24 h	180 °C	1458	2305	1.581
PLA-Sb_2_O_3_	1.0	24 h	180 °C	608	1120	1.843

## Data Availability

Data is contained within the article or Appendix A.

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
