# Peer review of "Microbial Synthesis of Lactic Acid from Cotton Stalk for Polylactic Acid Production"

_microorganisms, 2023, doi:10.3390/microorganisms11081931_

Round 1
Reviewer 1 Report
1. The introduction is very long, it is necessary to adjust it only with the most important things that impact the substantial results.
2. in the LAB isolation section, it should only describe the coordinates and the basics of sampling, in my opinion the figure is unnecessary as well as the table, even in the table it mixes methodology with results and it is incomprehensible, you should rewrite this section with the most necessary of the isolation.
3. Section 2.4 indicates that a biochemical characterization was performed and only presents a catalase test (this is insufficient since there are complementary biochemical tests) and mixes a description of the test that is unnecessary; only the methodology should be written in such a way that any reader can reproduce it.
4. Section Screening of Lactic Acid Bacteria, it is necessary to be congruent with the titles of the sections in this section indicates an screening, however, it seems to me that after describing a fermentation system, it should clarify and separate the sections as appropriate.
5. Line 190, page 6. If your intention is to describe the system you should provide all the details, for example what is your system (you should describe it, dimensions, geometry, are they flasks of what capacity, what type of lid they have, was a closed culture), when you indicate 10% of inoculum what do you mean, you should report the number of cells or CFU to which you adjusted the culture, mentioning percentages is not clear since 10% may contain different cell density, please clarify and justify adequately describing the system. Inclusively indicate that used an UV spectrophoto- 196 metric method at 390nm, How?
6. Line 223-224. Growth kinetics was 222 studied through estimating optical density every 6h interval using spectrophotometer at 223 600nm. How do you perform the analyses?
7. it is necessary to rewrite the sections 2.8.2 and 2.8.2. It is not clear, especially 2.8.2. which indicates that it is a chemical treatment to purify but starts by indicating that it is a chemical treatment to purify but starts by indicating that it adds cellulose. Clarify and clearly rewrite the sections
8. The sections do not have a correct order in the description of the methodology.
9. Line 332-336. The information is duplicated in table 3, you must decide where to describe the origin of the strains, in the table or in the text, it is not appropriate to duplicate the information.
10. In section 3.1 there is no discussion with the literature.
11. Figure 2. It would be important to put the image of the most representative strains that tested positive or, failing that, to put an image with the positive strains. You should also illustrate the image of your positive and negative control, there are, in the methodology you do not indicate which were your controls.
12. The results, it does not adequately describe the most relevant findings. It goes from a strain characterization to an analytical characterization, before a molecular one, where are the results of the fermentations, what is the time of maximum production, and which strains produce more, how do you argue this against the literature? What were the results of cellulose recovery, in general, I think the paper should be rewritten without omitting any information.
13. Figure 4 Figure 4 is of very poor quality
14. In the conclusions you should indicate the relevance of the work, you should not repeat results, you should draw the conclusion of this indicating the contribution of the work.
The wording should be revised
Author Response
Dear Reviewer,
We are thankful for analysing the manuscript minutely and help us in polishing the manuscript further to enable it getting the proper shape desired for its publication. Further, please find below the reply to comments as desired.
Reviewer 1 Comments
Comment 1. The introduction is very long, it is necessary to adjust it only with the most important things that impact the substantial results.
Reply to comment 1: We have incorporated desired changes in the introduction section in the revised manuscript.
Comment 2. In the LAB isolation section, it should only describe the coordinates and the basics of sampling, in my opinion the figure is unnecessary as well as the table, even in the table it mixes methodology with results and it is incomprehensible, you should rewrite this section with the most necessary of the isolation.
Reply to comment 2: The manuscript have been revised as per the reviewer’s suggestions.
Comment 3. Section 2.4 indicates that a biochemical characterization was performed and only presents a catalase test (this is insufficient since there are complementary biochemical tests) and mixes a description of the test that is unnecessary; only the methodology should be written in such a way that any reader can reproduce it.
Reply to comment 3: We have incorporated desired changes in the revised manuscript.
Comment 4. Section Screening of Lactic Acid Bacteria, it is necessary to be congruent with the titles of the sections in this section indicates an screening, however, it seems to me that after describing a fermentation system, it should clarify and separate the sections as appropriate.
Reply to comment 4: We have incorporated desired changes in the revised manuscript.
Comment 5. Line 190, page 6. If your intention is to describe the system you should provide all the details, for example what is your system (you should describe it, dimensions, geometry, are they flasks of what capacity, what type of lid they have, was a closed culture), when you indicate 10% of inoculum what do you mean, you should report the number of cells or CFU to which you adjusted the culture, mentioning percentages is not clear since 10% may contain different cell density, please clarify and justify adequately describing the system. Inclusively indicate that used an UV spectrophoto- 196 metric method at 390nm, How?
Reply to comment 5: We have incorporated desired changes in the revised manuscript.
Comment 6. Line 223-224. Growth kinetics was 222 studied through estimating optical density every 6h interval using spectrophotometer at 223 600nm. How do you perform the analyses?
Reply to comment 6: The manuscript have been revised as per the reviewer’s suggestions.
.
Comment 7. It is necessary to rewrite the sections 2.8.2 and 2.8.2. It is not clear, especially 2.8.2. which indicates that it is a chemical treatment to purify but starts by indicating that it is a chemical treatment to purify but starts by indicating that it adds cellulose. Clarify and clearly rewrite the sections.
Reply to comment 7: We have incorporated desired changes in the revised manuscript.
Comment 8. The sections do not have a correct order in the description of the methodology.
Reply to comment 8: We have revised the methodology section in the revised manuscript.
Comment 9. Line 332-336. The information is duplicated in table 3, you must decide where to describe the origin of the strains, in the table or in the text, it is not appropriate to duplicate the information.
Reply to comment 9: We have incorporated desired changes in the revised manuscript.
Comment 10. In section 3.1 there is no discussion with the literature.
Reply to comment 10: We have incorporated desired changes in the discussion section in the revised manuscript.
Comment 11. Figure 2. It would be important to put the image of the most representative strains that tested positive or, failing that, to put an image with the positive strains. You should also illustrate the image of your positive and negative control, there are, in the methodology you do not indicate which were your controls.
Reply to comment 11: The manuscript have been revised as per the reviewer’s suggestions.
Comment 12. The results, it does not adequately describe the most relevant findings. It goes from a strain characterization to an analytical characterization, before a molecular one, where are the results of the fermentations, what is the time of maximum production, and which strains produce more, how do you argue this against the literature? What were the results of cellulose recovery, in general, I think the paper should be rewritten without omitting any information.
Reply to comment 12: The manuscript have been revised as per the reviewer’s suggestions.
Comment 13. Figure 4 Figure 4 is of very poor quality.
Reply to comment 13: The manuscript have been revised as per the reviewer’s suggestions.
Comment 14. In the conclusions you should indicate the relevance of the work, you should not repeat results, you should draw the conclusion of this indicating the contribution of the work.
Reply to comment 14: We have incorporated desired changes in the revised manuscript.
Thank you.
Reviewer 2 Report
The authors have investigated the efficacy of cotton stalk as a potential product in agriculture that may serves as a beneficial, low-cost material as a medium for microbial synthesis of lactic acid as desired for polylactic acid synthesis. This study may be helpful to utilize microbial sources to counter heavy metal(s) stress on crops. The paper is generally well-written but the methodology framework is unclear and lacks potential references.
The introductory section of this article lacks critical and in-depth analysis of the available literature and comprehensive relevant work.
Comment 1: As it is a global issue, therefore, authors should incorporate quantitative data about cotton stalks in agriculture lands of different countries and it would be good to cite and relate some other studies that investigated similar aspects.
Comment 2: I suggest you add 2-3 introductory sentences about cotton stalks and microbial diversity in different regions of world. The reason to select cotton stalks as a testing media should be explained.
Comment 3: Explain the novelty of the work in a paragraph and make a comparison with the literature. There is a major gap to link previous work and the planned objectives of this work.
Comment 4: Besides the wide availability of cotton stalks, how about its relationship to native microbial communities?
Comment 5: Detailed information should be given about sample collection, preservation, and analytical techniques. There is no reference given in the materials and methods section. For microbial analysis cite below mentioned three references.
https://doi.org/10.1016/j.cej.2019.123674
https://doi.org/10.1016/j.envpol.2021.116587
https://doi.org/10.1016/j.biortech.2021.125330
Comment 7: I suggest you incorporate longitude and latitude data of sampling points and describe them in brief.
Comment 8: There are several citations in methods that should be replaced with primary references.
Comment 9: Authors should precisely focus on and strengthen this section with published research work.
Comment 10: I suggest you conduct a multivariate statistical analysis and determine the interaction effect among key parameters of this study.
Comment 11: In the discussion section, the ecological significance is not pointed out clearly by the authors.
Comment 12: In order to meet the requirements for publication, the quality of figures and tables should be improved.
Comment 13: Please rewrite this section with a prime focus on your key findings.
There are numerous low-level errors in the text (including but not limited to unit symbols, punctuation marks, font sizes, and paragraph structure…). Please check the format of the text throughout this manuscript and unified its pattern. Correcting these errors should not be the responsibility of the reviewers. Please check it carefully throughout this manuscript.
Author Response
Dear Reviewer,
We are thankful tfor analysing the manuscript minutely and help us in polishing the manuscript further to enable it getting the proper shape desired for its publication. Further, please find below the reply to comments as desired.
Comment 1: As it is a global issue, therefore, authors should incorporate quantitative data about cotton stalks in agriculture lands of different countries and it would be good to cite and relate some other studies that investigated similar aspects.
Reply to comment 1: We have incorporated desired changes in the revised manuscript.
Comment 2: I suggest you add 2-3 introductory sentences about cotton stalks and microbial diversity in different regions of world. The reason to select cotton stalks as a testing media should be explained.
Reply to comment 2: The manuscript have been revised as per the reviewer’s suggestions.
Comment 3: Explain the novelty of the work in a paragraph and make a comparison with the literature. There is a major gap to link previous work and the planned objectives of this work.
Reply to comment 3: We have incorporated desired changes in the revised manuscript.
Comment 4: Besides the wide availability of cotton stalks, how about its relationship to native microbial communities?
Reply to comment 4: We have incorporated desired changes in the revised manuscript.
Comment 5: Detailed information should be given about sample collection, preservation, and analytical techniques.
Reply to comment 5: We have revised the methodology section in the revised manuscript as per the reviewer’s suggestions.
There is no reference given in the materials and methods section. For microbial analysis cite below mentioned three references.
https://doi.org/10.1016/j.cej.2019.123674
https://doi.org/10.1016/j.envpol.2021.116587
https://doi.org/10.1016/j.biortech.2021.125330
Reply to comment 6: We have revised the methodology section in the revised manuscript as per the reviewer’s suggestions.
Comment 7: I suggest you incorporate longitude and latitude data of sampling points and describe them in brief.
Reply to comment 7: We have incorporated desired changes in the revised manuscript.
Comment 8: There are several citations in methods that should be replaced with primary references.
Reply to comment 8: The manuscript have been revised as per the reviewer’s suggestions.
Comment 9: Authors should precisely focus on and strengthen this section with published research work.
Reply to comment 9: We have incorporated desired changes in the revised manuscript.
Comment 10: I suggest you conduct a multivariate statistical analysis and determine the interaction effect among key parameters of this study.
Reply to comment 10: We have incorporated desired changes in the revised manuscript.
Comment 11: In the discussion section, the ecological significance is not pointed out clearly by the authors.
Reply to comment 11: We have revised the discussion section in the revised manuscript as per the reviewer’s suggestions.
Comment 12: In order to meet the requirements for publication, the quality of figures and tables should be improved.
Reply to comment 12: The manuscript have been revised as per the reviewer’s suggestions.
Comment 13: Please rewrite this section with a prime focus on your key findings.
Reply to comment 8: The manuscript have been revised as per the reviewer’s suggestions.
Thank you.
Reviewer 3 Report
Utilizing Cotton Stalk Waste for Sustainable Bioplastics Development - A Waste to Wealth Strategy
Manuscript ID: microorganisms - 2493862
Comments to the Authors
This paper addresses the problem of large volumes of biowaste and aims to reduce a particular biowaste, cotton stalk by using it as a substrate for synthesis of lactic acid (LA) which can be further used in to make polylactic acid (PLA), a sustainable bioplastic. The authors have reported the isolation, screening, and selection of LAB microbes used for LA synthesis. They have also presented the synthesis of PLA from LA using catalysts such as chitosan, yttrium stabilized zirconium (YSZ), and antimony trioxide, while optimizing other process parameters such as temperature, reaction time, and solvent (m-xylene).
While going through the manuscript, I realized that the authors have done an extensive amount of work, but the presentation can be improved. I feel no additional experiments are required but some major changes in the presentation of the manuscript is required to make it more readable and sound for the targeted audience. Please find my comments below:
1. In the current form, the title and abstract appears like the paper is a review paper. Because the paper addresses using cotton stalk for PLA, my suggestion would be to use those exact words in the title to make it sound like a research paper and not a generic review article.
2. I also feel that in the current form, the abstract and the manuscript lacks a strong conclusion. Although towards the end it has been briefly discussed the potentials of yttrium catalysts, there is no strong conclusive comment anywhere which states the benefits of the study or the conclusive results of the study.
3. The methods have been quite well described. I think it might also be helpful to add how the physicochemical parameters in table 1 were measured.
4. The manuscript has good figures but they have not been discussed very well in the text. I would find it very helpful if the discussions on figures are included in the main text. It might also be useful to have a little more descriptive figure legends.
5. The authors have very well stated the findings of their experiments but I feel the paper lacks a bit on what these results imply. It might be useful if the authors provide their thoughts on how their results connect to previous findings in literature and how these current findings might impact future directions of research.

Author Response
Dear Reviewer,
We are thankful for analysing the manuscript minutely and help us in polishing the manuscript further to enable it getting the proper shape desired for its publication. Further, please find below the reply to comments as desired.
While going through the manuscript, I realized that the authors have done an extensive amount of work, but the presentation can be improved. I feel no additional experiments are required but some major changes in the presentation of the manuscript is required to make it more readable and sound for the targeted audience. Please find my comments below:
Comment 1. In the current form, the title and abstract appears like the paper is a review paper. Because the paper addresses using cotton stalk for PLA, my suggestion would be to use those exact words in the title to make it sound like a research paper and not a generic review article.
Reply to comment 1: We have incorporated desired changes in the revised manuscript as per reviewer’s suggestions.
Comment 2. I also feel that in the current form, the abstract and the manuscript lacks a strong conclusion. Although towards the end it has been briefly discussed the potentials of yttrium catalysts, there is no strong conclusive comment anywhere which states the benefits of the study or the conclusive results of the study.
Reply to comment 2: The abstract and conclusion in the revised manuscript have been revised as per the reviewer’s suggestions.
Comment 3. The methods have been quite well described. I think it might also be helpful to add how the physicochemical parameters in table 1 were measured.
Reply to comment 3: We have incorporated desired changes in the revised manuscript as per reviewer’s suggestions.
Comment 4. The manuscript has good figures but they have not been discussed very well in the text. I would find it very helpful if the discussions on figures are included in the main text. It might also be useful to have a little more descriptive figure legends.
Reply to comment 4: We have incorporated desired changes in the revised manuscript.
Comment 5. The authors have very well stated the findings of their experiments but I feel the paper lacks a bit on what these results imply. It might be useful if the authors provide their thoughts on how their results connect to previous findings in literature and how these current findings might impact future directions of research.
Reply to comment 5: We have incorporated desired changes in the revised manuscript.
Thank you.
Reviewer 4 Report
In this manuscript, the authors investigated the isolation of lactic acid bacteria from natural sources to identify potential hydrolysates that can further utilize sugars present in lignocellulosic materials for the production of Lactic Acid. In my opinion, the manuscript is good, very easy to read and the authors presented a good amount of data. I think it is suitable for publication.
Author Response
Dear Reviewer,
thank you so much for your time and your positive feedback.
Round 2
Reviewer 1 Report
The paper can be accepted however figure 2 is not understood, labels are required for figures a, b, c.... and describe each one in the figure caption. This change is necessary otherwise the figure is unnecessary.
Author Response
Dear Reviewer,
the changes have been made.
Thank you again.